PATMA: parser of archival tissue microarray

Roszkowiak Lukasz lroszkowiak@ibib.waw.pl 1
Lopez Carlos clopezp.ebre.ics@gencat.cat 2
1 Laboratory of Processing Systems of Microscopic Image Information, Nalecz Institute of Biocybernetics and Biomedical Engineering, Polish Academy of Sciences , Warsaw , Poland
2 Molecular Biology & Research Laboratory (IISPV, URV), Hospital de Tortosa Verge de la Cinta , Tortosa , Spain
Tretiakova Maria
Electronic publication date: 2016 Dec 1
Publication date: 2016
Volume: 4
Electronic Location ID: e2741
Received 2016 Aug 29; Accepted 2016 Oct 31
Copyright: ©2016 Roszkowiak and Lopez
Copyright year: 2016
Copyright holder: Roszkowiak and Lopez
License: This is an open access article distributed under the terms of the Creative Commons Attribution License, which permits unrestricted use, distribution, reproduction and adaptation in any medium and for any purpose provided that it is properly attributed. For attribution, the original author(s), title, publication source (PeerJ) and either DOI or URL of the article must be cited.
License URL: https://creativecommons.org/licenses/by/4.0/

Keywords: Biomedical engineering, Virtual slide, Whole slide imaging, Tissue microarray, Image processing, Automatic segmentation, Image segmentation

Funding: European Union Project PO KL “Information technologies: research and their interdisciplinary applications” UDA-POKL.04.01.01-00-051/10-00 Institute of Health Carlos III PI11/0488 The study is cofounded by the European Union from resources of the European Social Fund. Project PO KL “Information technologies: Research and their interdisciplinary applications,” Agreement UDA-POKL.04.01.01-00-051/10-00. This work is cofounded by grant number PI11/0488 from the Institute of Health Carlos III (Instituto de Salud Carlos III), Spain. The funders had no role in study design, data collection and analysis, decision to publish, or preparation of the manuscript.

==============================
Tissue microarrays are commonly used in modern pathology for cancer tissue evaluation, as it is a very potent technique. Tissue microarray slides are often scanned to perform computer-aided histopathological analysis of the tissue cores. For processing the image, splitting the whole virtual slide into images of individual cores is required. The only way to distinguish cores corresponding to specimens in the tissue microarray is through their arrangement. Unfortunately, distinguishing the correct order of cores is not a trivial task as they are not labelled directly on the slide. The main aim of this study was to create a procedure capable of automatically finding and extracting cores from archival images of the tissue microarrays. This software supports the work of scientists who want to perform further image processing on single cores. The proposed method is an efficient and fast procedure, working in fully automatic or semi-automatic mode. A total of 89% of punches were correctly extracted with automatic selection. With an addition of manual correction, it is possible to fully prepare the whole slide image for extraction in 2 min per tissue microarray. The proposed technique requires minimum skill and time to parse big array of cores from tissue microarray whole slide image into individual core images.

Introduction

Background

In modern pathology, the use of tissue microarrays (TMA) is a common practice (Kallioniemi et al., 2001; Ilyas et al., 2013; Rexhepaj et al., 2013). TMA enables researchers to extract small cylinders of tissue, called cores or punches, from histological sections and arrange them in an array on a paraffin block so that multiple samples can be processed simultaneously. From this combined paraffin block slices can be cut and processed like any other histological section. TMA allows a more efficient way to analyse tissue sections, especially when they should undergo further processing, such as staining. It is possible to use immunohistochemistry (Neuman et al., 2013; Korzynska et al., 2010; Lopez et al., 2014), in situ hybridization (Garcia-Rojo, Bueno & Slodkowska, 2010), or fluorescence in situ hybridization (Du & Dua, 2010) on all cores simultaneously. This way, it lowers the intralaboratory variability of different staining concentrations between slices. Furthermore, it significantly improves processing time, allows obtaining more reproducible results, and greatly reduces the cost of analysis.

TMA are mostly used for cancer tissue evaluation. The analysis of TMA is very similar to regular analysis of histological slides (Seidal, Balaton & Battifora, 2001) as it is mostly based on visual examination, but instead of one specimen there are many in one slide. Nevertheless, reading big arrays of cores can be burdensome and requires caution. As the cores are not labelled directly on the slide with any text or number, the only way to distinguish specimens is based on their arrangement. The assumed grid structure (array) of the TMA with cores located on the grid nodes is often distorted through the mechanical procedures performed during the glass slide preparation (Fowler et al., 2011; Werner et al., 2000). It is very important to distinguish the correct order that was executed while the TMA was prepared.

Since computer-aided observation and analysis of tissue samples is increasingly common idea and TMA slides can be treated as normal histological slides, it is also possible to make their digital copy. Slide scanners create a virtual image of the whole glass slide (Kayser et al., 2006; Kayser et al., 2010; Pantanowitz et al., 2013). Whole slide images of TMA are enormous, in the context of size, pixels (resolution), as well as disk space. It is impossible to process these images at full resolution due to limited memory capabilities of a standard computer workstation. Splitting the whole virtual slide into separate images of individual cores is required for further image processing. The size of these images is suitable for analysis using a 64-bit based computer systems. Therefore, the image-processing algorithms (Redondo et al., 2012; Roszkowiak et al., 2016; Markiewicz et al., 2006; Neuman et al., 2010; Korzynska et al., 2013) process cores rather than whole slides.

Although TMA is gaining popularity in clinical practice, it is still more common in research facilities. A vital step to improve the speed and quality of this process is to correctly locate each tissue core in the array. Usually the tissue cores are aligned in a typical microarray, but quite often the TMA cores are askew and incomplete. The problems with automatic selection of cores in TMA include: places where the core should be located, but is actually absent because of technical issues; incomplete cores; cores consisting of many fragments; heavily skewed rows, where cores are relocated in unpredictable manner; stain deposits in the background.

Aim

The main aim of this study was to develop a computerized procedure capable of automatically finding and extracting cores from images of TMA. This software was developed on special request from Molecular Biology & Research Laboratory from Hospital Verge de la Cinta (Institut d’Investigacio Sanitaria Pere Virgili—IISPV), located in Tortosa, Spain. It was named PATMA for Parser of Archival TMA. The procedure is capable of handling core sections under standard conditions—moderately aligned array with well contrasted cores from the background. Once the TMA cores are located, they are stored in separate files for further assessment.

Related works

Currently, there is lack of available software that helps the researcher to extract single core image from the whole slide image, which is free of charge, simple to use, and automatic.

There are a few scientific works (Lahrmann et al., 2010; Dell’Anna et al., 2005) that can be related to this kind of image processing of TMA. Most of the software focuses on the designing (Thallinger et al., 2007) or scoring punches of TMA (Amaral et al., 2013). Some have been created to support specific study and do not support various data. Other programs apply manual processing (Chen, Reiss & Foran, 2004). The user must set the number of rows and columns and sometimes select the first core manually. Web-based solutions (Della Mea et al., 2006; Demichelis et al., 2006; Galizia et al., 2008) are not very efficient while working with high-resolution whole slide image, because transferring files with a size of 30 gigabytes or more is rather problematic. Some programs are meant to work with computer-controlled microscope for collecting data (Rabinovich et al., 2006); with addition of TMA planning program this may give good results for new research but not while handling archival data. Other solutions found in the literature include Image Miner (Foran et al., 2011), recently published complex computer-aided diagnosis system (Milagro Fernandez-Carrobles et al., 2014; Milagro Fernandez-Carrobles et al., 2015) for TMA, and others (Morgan et al., 2004; Liu et al., 2005).

There are also commercial solutions such as TMALab from APERIO or TMADesigner2 from APHELYS. The major disadvantage of commercial software is that it is bound to a specific slide scanner.

Method

The software was developed in MATLAB R2014b implemented on Intel Core i7-4710MQ @2.50 GHz CPU, 16.0GB RAM. The created procedure consists of the steps presented in Fig. 1.

File reading

Virtual slide files usually have an inner structure of tiled images that can form the full picture in multiple resolutions. Loaded files are investigated for available resolutions concealed inside. Depending on the user’s choice—manual or automatic—working resolution is selected for analysis. Automatic selection always chooses the smallest resolution. Manual selection gives user a choice by providing all available resolutions. The selected image is then preprocessed. As of now, the supported format for virtual slides is single-file pyramidal tiled TIFF file format.

Figure 1 Software procedure.

Preprocessing

The aim of this preprocessing is to create a binary mask with punch location. First, the image is converted from RGB to Lab colour space. Image is binarised with threshold value defined as most common intensity in Luminescence layer (Lab colour space) decreased by 5. Then, the morphological area opening is executed. Depending on the selected working resolution, objects containing fewer than 1,000 pixels are removed from images smaller than 107 square pixels and objects fewer than 10,000 pixels from larger images. Next, a series of other morphological operations including closing, filling, and border clearing are performed (see Algorithm 1). As a result of the preprocessing we obtain black and white images where objects correspond to punch (or fragment of punch) location.

Occasionally, strong stain deposit in the background or overstained tissue sections cause extremely unsatisfactory results. In such case, user can choose expert mode and manually modify threshold value as well as size limit of the object to be treated as punch and size of structural element used for morphological operations. If the TMA image was loaded with a wrong orientation, the possibility of rotating the whole TMA is also available in expert mode section. Through this expert preprocessing, a binary map can be previewed by the user to adjust the parameter values to a particular image.

Punch handling

We collect properties of found objects (area, centroid, eccentricity, and bounding box—standard MATLAB features). Then, we try to assess the common distance between the cores. This reference distance (refDist) is calculated based on the most circular punch (object with the minimum value of eccentricity) as halved mean of bounding box’s width and height. We assume that the distance between the punches is no greater than reference distance. We also assume that punches are objects with eccentricity lower than 0.9. Objects that fulfill the criteria are assigned to columns and rows based on their centroid coordinates (see Algorithm 2).

First, objects are assigned to columns one after the other. Among all the unassigned objects we seek the one with the smallest horizontal coordinate. Then, we look for objects with centroid located not further than reference distance, but only in terms of horizontal coordinate. The valid objects are assigned to currently evaluated column. Then, we look for objects in the next column.

After objects are allocated to columns, we try to determine the proper row assignment. First step is similar to that in column search; the object with the lowest vertical coordinate is found. Then, we examine the rest of the objects if they are valid for this row. Objects within the reference distance in terms of vertical coordinate are assigned to currently evaluated row. If this criterion is not met, we have an additional condition based on the centroids of objects already assigned to the currently examined row. We evaluate the polynomial of second degree based on at least three points. Then, we make an extrapolation of that polynomial with the horizontal value of the currently examined centroid. If the assumed point is within the reference distance of examined centroid, then it is also assigned to the current row. Example of such assignment is shown in Fig. 2.

Figure 2 Example of object assignment to row.

Centroids of objects are marked with green rectangles. Horizontal blue solid line has originated from the centroid of the first object in currently evaluated row. Dashed blue lines mark the reference distance. If the centroid of another object is within the reference distance, it is assigned to the currently evaluated row. In this example, six objects (outside gray border in (B)) meet this term—are within reference distance. The next object’s centroid is outside the zone of reference distance, so it would be not assigned to this row. In (A) there is a polynominal (red line) of second degree, based on the centroids of already assigned objects, with one extrapolated point (yellow triangle). Extrapolated point has the same horizontal value as the centroid of the next object. Because extrapolated point is closer to the centroid of the object than the reference distance, it is assigned to the currently evaluated row. (B) shows final result for this row with three polynominals evaluated for all objects outside of the reference distance from assumed horizontal line. (C) is an enlarged fragment of image (B) within gray border.

Then, objects with a missing value of row (or column) are managed. The distance between evaluated object’s centroid and the nearest object’s centroid is calculated. If it is greater than the reference distance, analyzed object is assigned to the adjacent row (or column). However, for objects closer than the reference distance, closest objects are combined/merged into one. Whenever object has no row and column value, a mean of coordinates of each column and row is calculated. Then, this object is assigned to the closest row and column by distance.

Finally, duplicate assignments to rows and columns are treated with the same algorithm. Around every object a region of interest (ROI) similar to the bounding box is created. The regions are adjusted so that there is no overlap. Lastly, correct numerical labels, corresponding to TMA punch numbering, are applied to each ROI.

The designed array structure of the TMA is often distorted, cores can be relocated and may also disappear completely leaving “holes.” Because we evaluate assumed reference distance between the cores of TMA, we can figure out where the cores should be in the array despite their actual absence. We assign successive numerical label to the identified “holes” (ROI object with turned off Validity option—see the following paragraph).

User input

The ROI selected in the previous step are presented as an overlay in the processed image, as in Fig. 3. At this stage, the user may apply any necessary corrections. Moving and resizing of selected ROI is possible, because it is draggable. Manual corrections of numerical labels can be applied. Moreover, there is an option of addition and deletion of ROI.

Figure 3 Graphical user interface of the developed software.

TMA with overlay of regions of interest presented in the main window. This interface also contains subsections related to the file selection, preprocessing, the list of ROI, and extraction.

To make editing easier, there is a table presented in the GUI that consists of Numerical label, Position parameters, and Validity. Validity option provides a way to keep correct numbering of punches while there are empty spaces between them. Objects marked as not valid are not extracted in the final step. All of these attributes can be manually changed.

Extraction

The selected ROI containing punches are extracted from the maximum available resolution image as separate files. The user can add border margin to the extracted punch to be sure that every pixel with information is inside the selected ROI. Only one punch or all punches from the TMA can be extracted. Images are saved with easily distinguishable filenames in TIFF format.

Results

There were 27 virtual slides of TMA available to perform the evaluation of this software. The files were received from Molecular Biology & Research Laboratory from Hospital Verge de la Cinta (IISPV). The glass TMA slides have been prepared according to the procedure described in detail in Lopez et al. (2014). The WSI has been acquired using the whole slide scanner (ScaneScope, Aperio) with 40×/0.75 Plan Apo objective lens. We used all the available data with the exception of one file that had to be excluded, as the image file contained only a part of the whole TMA which was also rotated. Results are presented in Table 1. In five cases we used resolution chosen manually, as it yielded much better results than automatically selected minimal available resolution (in pyramidal tiled TIFF file). In 11 cases we had to use expert mode (see Preprocessing in the Methodology section) to improve the results, due to strong stain deposit in the background. About 58% of images were processed by entirely automatic method.

Table 1 All results of core selection on 26 valid cases.

Approximate values of segmented miniature image size are presented in Resolution column. Wherever higher resolution was chosen manually, as it yielded much better results than minimal available resolution, there is higher res comment. Punches in TMA are cores planned in the TMA (regardless if the core is still in the image). Selected objects is the number of correctly set ROI (including cores and “holes”). Erroneous objects are all extra, unnecessary ROI set in the image that have to be deleted manually.

Case	Resolution (approx.)	Punches in TMA	Selected object	Erroneous object	Comments	
1	3,800 × 2,700	50	50	4		
2	4,000 × 2,700	50	48	5		
3	7,300 × 5,000	50	46	1	Higher res	
4	3,800 × 2,700	50	44	24		
5	3,900 × 2,700	50	45	1		
6	3,800 × 2,700	50	45	40		
7	7,800 × 3,900	40	38	14	Higher res, expert mode	
8	3,800 × 2,000	40	38	12	Expert mode	
9	3,700 × 2,300	40	35	12	Expert mode	
10	3,600 × 1,900	40	30	0	Expert mode	
11	7,300 × 4,500	50	45	10	Higher res	
12	2,500 × 1,300	50	40	8	Expert mode, askew	
13	8,700 × 5,200	50	47	4	Higher res	
14	7,200 × 5,200	50	39	4	Higher res, askew	
15	3,400 × 2,400	50	45	29		
16	3,500 × 2,400	50	48	7		
17	3,600 × 2,600	50	43	7		
18	3,500 × 2,300	50	45	7		
19	3,500 × 2,300	50	49	7		
20	3,600 × 2,600	50	39	4	Expert mode	
21	2,000 × 1,300	50	41	5	Expert mode, askew	
22	3,600 × 1,800	32	32	6		
23	3,600 × 2,300	50	45	1	Expert mode	
24	2,000 × 1,300	45	36	4	Expert mode	
25	3,300 × 1,400	32	27	1	Expert mode	
26	3,700 × 2,400	50	45	2	Expert mode	
	SUM	1,219	1,085	219		

The number of punches per virtual slide varied from 32 to 50. We assume that satisfactory selection consist of ROI with punch. The total percentage of correctly selected cores in 26 evaluated TMA was 89%. In total, 1,085 out of 1,219 punches were selected correctly as a whole core with index of correct row and column. In the majority (16) of cases we achieved 90% or more correctly selected punches in the TMA slide. In only seven images PATMA method resulted in selecting over 10 erroneous objects. The number of objects which were incorrectly assigned can be most definitely lowered with human interaction (expert mode).

The PATMA’s semi-automatic processing approximately takes less than 2 min to fully prepare the TMA slide if the quality of TMA and scanned image is satisfactory. For images that need more human interaction takes about 5 min.

Discussion

The processing of fully automatic method is not supervised by the user, hence it may cause unavoidable errors in the extracted images. Eventually, it may lead to additional time-consuming work (manual task) if punches are extracted with wrong labels (numerical label of punch defines its location in TMA, hence the specimen identification), fragments of punches are extracted separately or punches that are touching border of the image. Thus, we decided to give an option of user supervision in the proposed method.

PATMA identifies not only cores but also “holes”—places where the core should be located, but is actually absent because of technical issues. As a result, we can assign successive numerical labels to all locations of designed array structure of the TMA. It is very useful to identify the “holes” to keep the correct numerical labelling of cores. It is critical to distinguish the correct order that was executed while the TMA was formed as this is the only way to distinguish specimens.

It takes less than 2 min to process a good quality TMA slide image with PATMA and for images that need more human interaction it takes about 5 min. However, complete manual processing is much more laborious and prone to error. Processing the TMA images in full resolution is impossible, for that reason they have to be opened in pieces or the coordinates from their miniatures have to be manually converted to full resolution. It is possible to perform, for example in Fiji (ImageJ), but it would take a lot of effort and time. For the purpose of comparison we performed manual cores extraction on single TMA image with 50 cores and this took about 5 h.

One of the major problem we encountered are heavily skewed rows of TMA. Relocation of punches often happens during many steps of the TMA slide formation (Korzynska et al., 2014; Yaziji & Barry, 2006). It is impossible to predict if the punches will form askew line or will be randomly relocated. Sometimes two consecutive lines go in different directions in the final image. In this type of deformation it is hard to determine the correct row of the punch even for a human expert. We achieved our goal in case of this problem; from 3 TMA with most askew rows (see Table 1), the 80% of cores have been correctly selected without human interaction.

Another problem is that the staining of TMA is often not carried out perfectly, which causes deposits and overstained tissue parts in the background of virtual slide (Stromberg et al., 2007; Lejeune et al., 2013; Liu et al., 2002). This may result in segmentation of erroneous objects. In 7 images, where PATMA selected over 10 erroneous objects, it was due to stain deposit. Expert mode should be used whenever there is a stain deposit in the background to handle that problem manually.

Incomplete cores or those with many fragments are not a great problem. They usually are well located and merged into one ROI. Artifacts present in the image are much more difficult to handle. Big glue deposit is usually treated as an additional punch or merged with the closest punch. It is a deliberate procedure, because it is easier to delete additional ROI than to precisely create a new one. For the slides with extra glue or stain deposits or with cores consisting of many fragments our strategy to remove manually all extra artifacts appears to be efficient. It takes a very small amount of operator’s time to make adjustments; the whole procedure time takes up to 2 min.

In some cases, the amount of corrections that were necessary may seem high, but they are performed quickly and take much less time than extracting single punches manually. There are 219 erroneous objects (false positives) in evaluated dataset but they can be effortlessly discarded during user inspection. Handling of erroneous object takes no more than 2 s. However, 134 (11%) cores were not found (false negatives) in the evaluated dataset and should be added manually. Handling missing objects consist of adding new ROI, moving it to correct location in the TMA and adjusting the borders. It takes more time and effort than deleting erroneous object but with use of PATMA it is still faster and more straightforward than manual extraction. In total, it takes 2–5 min to set up all necessary ROI in TMA whole slide image for extraction. In comparison with fully manual processing, which takes hours, the advantage of PATMA is beyond doubt.

Furthermore, we encountered a problem to evaluate our method in comparison to any other related work. None of the related works describe their methodology in detail, hence their solutions (algorithms) are irreproducible. We also tried to contact most of the authors of related works without success. None of the related software was obtainable, as they are only theoretically available on the web. Lastly, the great advantage will be actual availability of the software as we want to share it through MathWorks File Exchange and update it in the future.

Conclusions

We present free software to efficiently extract separate punches from virtual slides of TMA. It is simple to use for any pathology laboratory, molecular research laboratory, or any other research facility. The great advantages of our software are that it requires none to minimal user input for typical TMA virtual slides and minimum skill and time to operate.

This automation is an advantage for the Molecular Biology & Research Laboratory compared to the previously used software, which was fully automatic, but worked without considering the location of cores in rows and columns. The time needed to relabel cores so they would match correct specimen was considerably longer than the proposed semi-automatic processing. The use of PATMA allows a substantial reduction in the processing time and easier error handling.

Future work

In the future, we want to further develop the software to improve its performance. We plan to implement the background elimination algorithm in the preprocessing stage to cope with strong stain deposits. We would like to improve punches’ assignment to correct rows and columns, even in heavily skewed tissue microarrays, for example by assuming grid structure that would be matched elastically to actual structure. Finally, we plan to expand the supported file types, so that the other file formats would be handled.

Supplemental Information

Supplemental Information 1 PATMA: Parser of archival tissue microarray

PATMA software source code created and tested in MATLAB R2015b.

Click here for additional data file.

Additional Information and Declarations

Competing Interests

Author Contributions

Data Availability

The authors declare there are no competing interests.

Lukasz Roszkowiak conceived and designed the experiments, analyzed the data, wrote the paper, prepared figures and/or tables, reviewed drafts of the paper, designed and developed the software.

Carlos Lopez conceived and designed the experiments, contributed reagents/materials/analysis tools, reviewed drafts of the paper, discussed the methodology.

The following information was supplied regarding data availability:

MathWorks: http://www.mathworks.com/matlabcentral/fileexchange/60131-patma--parser-of-archival-tma?requestedDomain=www.mathworks.com.

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
