# Peer review of "PATMA: parser of archival tissue microarray"

_PeerJ, doi:10.7717/peerj.2741_

## Round 0.1 · original submission · Minor Revisions

· Academic Editor

Minor Revisions

TMA is a great methodolgy and resource for research community and tissue based studies/clinical trials. This study contributes to ease of TMA utilization and promotes accuracy, speed and quality of TMA interpretation and analysis. It was surprising, however, that all TMA slides utilized for software analysis contained approximately 50 cores, which is way below average number of cores on TMA. High-throughput TMA methodology refers for hundreds of cores, not dozens. Usually, each case/specimen is arrayed in triplicate, therefore 50 cores could represent only about 15 cases with some controls added. Another concern is that given examples represent the simplest possible mapping scheme.

Therefore, it will be important to 1) add analysis of larger TMA (at least ~100 cores) and 2) more complex TMA maps clustering cases into sectors/groups with empty rows and columns between (very common arrangement in TMA design).

Reviewer 1 ·

Basic reporting

No Comments

Experimental design

No Comments

Validity of the findings

No Comments

Additional comments

The authors present an in-house developed, free software which can semi-automatically catalog and position tissue microarray (TMA) cores from standard TMA slide preparations, thus helping in cataloging, digital storage of images, easier navigation, as well as easier retrieval and comparison e.g. when multiple immunohistochemical stains are performed on a TMA for a research study. The authors have made their algorithm available, which can be of great aid for other researchers who might attempt modifications or improvements. Despite some limitations that the method has, this can be a very useful tool for research, especially given its availability as freeware.
Comments to the authors are found below:
1. 89% accuracy makes for the need of a semi-automated approach. The authors make a point of underlining this in the manuscript; however, it would be useful if they elaborated on the manual part of the method e.g. pointing out the need for a Pathologist to evaluate the slide for correct assignment of cores.
2. Page 2, row 32: Typographical error "specimen" instead of "specimens". Please rephrase.
3. The authors state: "The assumed grid structure (array) of the TMA with cores located on the grid nodes is often distorted because of elasticity of paraffin". Please provide either a visual (photo or schematic) representation and/or provide reference(s) for this statement. Also, it would be interesting to see if any data are available circa the bias that different TMA preparation methods may introduce.
4. The authors state: "Whole slide images of TMA are enormous, in context of size, in pixels (resolution) as well as disk space. It is impossible to process these images at full resolution due to limited memory capabilities of standard computer workstation". Whole-slide imaging is currently widely used on a clinical level in many areas of Pathology. Current end-user and server support is sufficient for this purpose, and it is standard of care for traditional histological preparation. Although mentioned in the manuscript, the authors should further explain how TMA slide imaging is different from conventional histological whole-slide imaging in this regard, if it actually is. The last sentence in the above statement should be checked for accuracy, with regard to the current state of Pathology Informatics.
5. Page 3, row 56: "automatically" instead of "automatic".
6. The limitations of the method are outlined by the authors; however, there should be a paragraph mentioning how these limitations are planned to be addressed.

Reviewer 2 ·

Basic reporting

The article is well organized, motivated an described. The sentence structure, however, requires substantial editing. The lack of articles and improper tense makes the presentation cumbersome. E.g., "We assign successive numerical labels to found holes", or "One of major problem we encountered are heavily skewed rows of TMA". A careful editing by a proofreader is advised.

Experimental design

The methods are clearly laid out and describe the utility of the authors' software.

Validity of the findings

The problem the authors address is a very practical one, yet one whose performance is difficult to evaluate. It appears to be fairly fast and easy, but the ultimate utility must be determined by the end users and their evaluation of its speed, ease and results with large and post-curated sets of TMA images.

Additional comments

no comments

---

## Round 0.2 · accepted · Accept

· Academic Editor

Accept

Dear Lukasz,

We found the revisions of your manuscript acceptable. Congratulations and thanks for your scientific contribution.

Dr Maria Tretiakova